# Multi-Sensor Scheduling Method Based on Joint Risk Assessment with Variable Weight

**DOI:** 10.3390/e24091315

**Published:** 2022-09-19

**Authors:** Lin Zhou, Jiawei Wu, Qian Wei, Wentao Shi, Yong Jin

**Affiliations:** 1School of Artificial Intelligence, Henan University, Zhengzhou 450046, China; 2School of Marine Science and Technology, Northwestern Polytechnical University, Xi’an 710072, China

**Keywords:** multi-sensor scheduling, risk assessment, multi-step prediction, convex optimization

## Abstract

In multi-sensor cooperative detection systems, to reduce target threat risk caused by attack tasks and target loss risk induced by uncertain environmental factors, this paper proposes a multi-sensor scheduling method based on joint risk assessment with variable weight. Firstly, considering the target state and prior expert experience of sensor scheduling, this paper gives a new scheme of target threat risk. Then, by combining the given target threat risk and the target loss risk, this paper constructs a joint risk model to meet the diversity of risk assessment. Secondly, a variable-weighted joint risk assessment model is given based on the adaptive weight of target loss risk and target threat risk, and the optimization problem of multi-sensor scheduling is described to minimize the multi-step prediction of the variable-weighted joint risk model. Finally, this paper relaxes above the non-convex optimization problem as a subconvex problem and designs the scheme of multi-sensor scheduling, improving the rapidity and optimization of the sensor scheduling solution. The simulation results show that the proposed method can adaptively schedule sensors and accurately track targets by using minimum sensor resources.

## 1. Introduction

The situation assessment of the multi-sensor cooperative system plays an important role in national defense security. In a complex dynamic environment, scheduling limited sensor resources to achieve accurate measurement of targets is important. Therefore, some sensor scheduling methods are proposed to optimize the performance of cooperative systems by scheduling sensors and assigning working parameters of sensors [1,2,3,4].

To effectively schedule sensors, some algorithms based on information theory are proposed in which information entropy was employed to evaluate the performance of sensor scheduling [5,6]. Instead of using information theory, Wang et al. proposed the linear programming method, which models sensor scheduling as an optimization problem composed by tracking accuracy and matching a matrix of sensors [7]. Without considering target tracking accuracy in the proposed scheme, the above methods experience poor detection performance. To enhance sensor tracking accuracy, a multi-sensor scheduling method based on PCRLB and interception probability factor was proposed in [8], where interception probability of the sensor was controlled within the threshold domain. However, since the objective function in the model was nonconvex, directly employing the gradient-descent method to tackle it was impracticable. As a part of the target tracking system, threat assessment is very important, it directly affects the scheduling of sensors, leading to the performance of target tracking decrease. Therefore, it is necessary to study threat assessment in the scheduling of sensors [9,10]. Unfortunately, the above methods are not involved.

To cope with the impact of risk assessment on sensor scheduling, Lan et al. proposed a scheduling method to switch sensors based on a target threat risk assessment [11]. However, Lan’s method merely focuses on a target threat risk, while ignoring the target loss risk. Pang et al. proposed a comprehensive risk assessment model firstly, in which target loss risk and sensor radiation risk are all considered, and then designed a scheme of sensor scheduling related to this model [12]. Since the ratio between target loss risk and sensor radiation risk are fixed, Pang’s method suffers performance deterioration at the ratio between target loss risk and sensor radiation risk changing. Mou et al. proposed a method to minimize target threat risk, in which an objective function of sensor scheduling was constructed by constraining higher sensor tracking accuracy and lower target threat risk [13]. Since an exhaustive search method is employed, Mou’s method suffers a heavy computational burden. Additionally, some multi-sensor scheduling methods have also been introduced to schedule sensors. Ebenezer et al. randomly assigned sensors to detect and track targets in [14], which solved the problem of multi-target tracking at a low signal-to-noise ratio (SNR). In [15], a closest scheduling method was provided, in which tracking accuracy and interception risk were considered for multi-sensor scheduling method, but selected sensors closer to the target.

So far, risk prediction is rarely involved in the above studies; thus, risk assessment performance may degrade with target maneuvering or environment changing. To cope with the problem, a myopic multi-sensor scheduling method was further introduced to reduce the target tracking risk in [16]. However, this method regards the minimization of one-step risk prediction as in the optimization rule to schedule sensors, where multi-step prediction is not be considered.

Obviously, we can find from the above analysis that target risk is single in traditional sensor scheduling methods; furthermore, the operating mode of the sensor can frequently be switched with risk change in a complex environment.

In this paper, we present a new multi-sensor scheduling method to tackle problems, including the multiple type risks, the computational burden and risk prediction. Compared with other existing methods, our method can not only combine multiple risks but also enhance multi-step prediction of risks. The main contributions are as follows.

(1)Joint risk assessment model containing target loss risk and target threat risk with variable weight is proposed. The traditional risk assessment methods only consider single risk (for example, target threat risk), while the proposed joint risk assessment model may be suitable for describing discrete risk and continuous risk, satisfying diversity of risk assessment models.(2)A multi-step joint risk prediction model is introduced to smooth the vibration of risk assessment, thus frequently switching the work mode of the sensor induced by time-varying risk can be avoided, and improve the optimization solution of sensor scheduling.(3)Convex relaxation is introduced to handle the NP-hard problem of sensor scheduling, leading to a suboptimal scheduling scheme with relatively minor computational complexity, which rapidity improves the algorithm.

The rest of this paper is organized as follows. Section 2 introduces the research scene and system model. Section 3 gives the variable weighted joint risk assessment model. Furthermore, the optimization problem of multi-sensor scheduling based on joint risk prediction assessment model is explored, and details of the proposed algorithm are presented in Section 4. Section 5 discusses the feasibility and effectiveness of the proposed algorithm. Section 6 provides some conclusions of this paper.

## 2. Problem Description

A multi-sensor cooperative detection system is composed of *N* sensors and *M* air targets. For the sake of simplicity, a two-dimensional model is considered in this paper; however, our method can be easily extended to three-dimensional scenarios. In Figure 1, every sensor can measure all targets by probe links and transmit these measurements to the processing center (PC) by communication links. In addition, the PC can send sensor measurements to the control center (CC) by communication links. Finally, the sensor scheduling scheme can be obtained according to the received measurements in CC, and then they are broadcasted to every sensor by communication links.

Some constraints are described as follows:(1)The number *N* of sensors used to detect targets is greater than the number *M* of targets, i.e., N>M.(2)One target can be tracked by at least one sensor.

According to the above scenario, we state the model of the moving target as:(1)$$x_{k}=\Phi x_{k-1}+\omega_{k-1}$$
where $x_{k}$ and $x_{k-1}$ are the target state at the moment *k* and k−1, respectively. $\Phi =\left[\begin{array}{ccc} I & A & B\\ O & I & A\\ O & O & I \end{array}\right]$ is the transition matrix of target state, I is the unit matrix size of J×J (*J* is the dimension of target state), A=T×I (*T* is the sampling period), $B=T^{2}I/2$, $O=0_{J\times J}$ is zero matrix. $\omega_{k-1}$ is Gaussian white noise with mean $0_{1\times J}$ and variance $Q_{k-1}=E\left[\omega_{k-1}\omega_{k-1}^{T}\right]$.

At the same time, the measurement model of the sensor is:(2)$$z_{k}=h\left(x_{k}\right)+\nu_{k}^{}$$
where $h\left(x_{k}\right)$ is the measurement matrix, $\nu_{k}$ is the Gaussian white noise with mean $0_{1\times \eta }$(η is the dimension of sensor measurement) and variance $R_{k}=E\left[\nu_{k}\nu_{k}^{T}\right]$. Furthermore, the measurement noise $\nu_{k}$ and the state noise $\omega_{k}$ are independent of each other.

## 3. Target Loss Risk and Target Threat Risk

The limited detection accuracy and capability of the sensor, lead to target loss. Additionally, some targets have weapon threat system security. To this end, we define risk as the product of the loss caused by the uncertain event, and the probability of the event. Thus, in this paper, risk mainly includes target loss risk caused by uncertain environmental factors, and target threat risk caused by attack tasks.

### 3.1. Target Loss Risk

Detection probability of the target is mainly determined by some factors, including sensor radiation power, signal-to-noise ratio, radiation time and the distance from sensor to target [17]. Therefore, target loss risk is constructed as:(3)$$W=\left(1-P_{D}\right)\cdot \sum_{r=1}^{n_{r}}\varepsilon_{r}th_{r}$$
where $P_{D}$ is the detection probability of the target; thus, $\left(1-P_{D}\right)$ can be regarded as the loss probability of the target. $n_{r}$ is the number of target attack indicators. $th_{r}$ is the target attack indicators. $\varepsilon_{r}$ is the weight of target attack indicators. Based on the scenario we researched, $n_{r}=2$ is given, and $th_{1}$ and $th_{2}$ are the nearest distance and nearest time of the target attack, respectively, while $\varepsilon_{1}$ and $\varepsilon_{1}$ are their weights.


(1)Detection probability of target $P_{D}$


Given the false alarm probability $P_{fa}$ of the electromagnetic detection system, the detection probability $P_{D}$ can be obtained as:(4)$$P_{D}=\frac{1}{2}erfc\left(\sqrt{-\ln P_{fa}}-\sqrt{SNR+0.5}\right)$$
where $erfc\left(\cdot \right)$ is a complementary error function.

Further, the signal-to-noise ratio (SNR) of the electromagnetic detection system can be described as [14]:(5)SNR=PacPacNnoiNFiNnoiNFi=PacPacKeT0BRiNFiKeT0BRiNFi
where Nnoi=11KeT0BRiKeT0BRi is the system noise, $K_{e}$ is the Boltzmann constant, $T_{0}$ is the noise temperature, $B_{Ri}$ is the receiver bandwidth. $N_{Fi}$ is the noise factor. $P_{ac}$ is the radiation power of the signal received, and it is expressed as:(6)Pac=PpeGTGRλ2GIPPpeGTGRλ2GIP4π2RD24π2RD2
where $P_{pe}$ is the peak power of the radiation pulse, $G_{T}$ is the transmitting antenna gain of the sensor, $G_{R}$ is the receiving gain in the target direction, λ is the sensor working wavelength, $G_{IP}$ is the net gain of the receiver processor, $R_{D}$ is the distance between the sensor platform and target.

(2) Loss related to target missing 

The loss of detection system caused by target missing, in which the nearest distance $th_{1}$ and the nearest time $th_{2}$ of target attack are analyzed, and then they can be described as [18]:(7)th1=1,dCPA≤d11−2dCPA−d1dCPA−d1d1−d0.5d1−d0.52,d1<dCPA≤d0.52dCPA−d0.5dCPA−d0.5d0.5−d0d0.5−d02,d0.5<dCPA≤d00,d0<dCPA
(8)th2=1,tCPA≤t11−2tCPA−t1tCPA−t1t1−t0.5t1−t0.52,t1<tCPA≤t0.52tCPA−t0.5tCPA−t0.5t0.5−t0t0.5−t02,t0.5<tCPA≤t00,t0<tCPA
where $d_{i}$ and $t_{i}$ represent the nearest distance target and the nearest time target when the loss value is *i*(i=0,0.5,1). Additionally, $d_{CPA}$ and $t_{CPA}$ can be expressed as:(9)tCPA=−(posx·velx+posy·vely+posz·velz)velx+posyvely+poszvelz)velx2+vely2+velz2velx2+vely2+velz2dCPA=posx+tCPA·velx2+posy+tCPA·vely2+posz+tCPA·velz2posxposyposzT=xkykzkT−x0y0z0TvelxvelyvelzT=x˙ky˙kz˙kT−x˙0y˙0z˙0T
where $pos_{x},pos_{y},pos_{z}$ and $vel_{x},vel_{y},vel_{z}$ represent the relative position and relative velocity of the *x*, *y* and *z* axis, $x_{k},y_{k},z_{k}$ and ${\dot{x}}_{k},{\dot{y}}_{k},{\dot{z}}_{k}$ represent the distance and velocity of the *x*, *y* and *z* axis, $x_{0},y_{0},z_{0}$ and ${\dot{x}}_{0},{\dot{y}}_{0},{\dot{z}}_{0}$ are the initial distance and velocity of a moving target, respectively.

### 3.2. Target Threat Risk

Comprehensively evaluating target threat risk from many aspects mainly includes two types, i.e., discrete risk and continuous risk. Discrete risk includes airborne weapons, jamming capability, category of target, and so on. Additionally, continuous risk includes heading angle, height, speed, distance, and so on. Furthermore, the above risks can be described in Figure 2.

To evaluate the target threat risk, we comprehensively consider discrete and continuous risks as:(10)$$Q=\sum_{j=1}^{m}f_{j}\left(U\right)\mu_{j}$$
where *U* is the target threat value, $f_{j}\left(U\right)$ is the probability of target threat risk level *j*, $\mu_{j}$ is the loss caused by target attack with threat risk level *j*. More details related to (10) can be introduced as follows.


(1)The target threat value *U*


By normalizing continuous risk related to target threat risk, they can be used in (10), and then normalized functions are described in Table 1.

In Table 1, $f_{C}$,$f_{A}$,$f_{V}$ and $f_{R}$ are the coefficient of heading angle, height, speed and distance, respectively. $U^{C}$, $U^{A}$, $U^{V}$ and $U^{R}$ are the heading angle, height, speed of target and distance between target and sensor, respectively. At the same time, $U_{\max}^{A}$, $U_{\max}^{V}$ and $U_{\max}^{R}$ are the maximum value of given variables, respectively.

Therefore, the target threat value based on Table 1 can be expressed as:(11)$$U=\phi_{C}C+\phi_{A}A+\phi_{V}V+\phi_{R}R$$
where $\phi_{C},\phi_{A},\phi_{V},\phi_{R}$ are the weights of heading angle, height, speed and distance, respectively.


(2)Probability of target threat risk $f\left(U\right)$


Given the fact that the probability of the target threat risk is discrete, it is necessary to discretize the continuous value *U* to calculate the probability of the target threat risk. For the sake of simplicity, we select the target threat value *U* at the integer moment to describe more details.

The probability of the target threat risk is defined as $p=\left[f_{1}\left(U\right),f_{2}\left(U\right),\cdots ,f_{m}\left(U\right)\right]$, and $f_{j}\left(U\right)\left(j=1,2,\cdots ,m\right)$ is the *j*-th level probability of the target threat. Additionally, the probability of the target threat at the moment *k* is:(12)$$p_{k}=p_{k-1}\cdot G_{k}$$
where $p_{k-1}$ is the probability of the target threat risk at the moment k−1, $G_{k}$ is the probability of the Markov transition at the moment *k*, and can be expressed as [19]:(13)$$\begin{array}{c} \begin{array}{c} G_{k}=\left[\begin{array}{ccccc} G_{11}-\lambda_{1}\left(k\right) & G_{12}+\lambda_{1}\left(k\right) & 0 & \cdots & 0\\ G_{21}-\frac{\lambda_{2}\left(k\right)}{2} & G_{22}+\lambda_{2}\left(k\right) & G_{23}-\frac{\lambda_{2}\left(k\right)}{2} & \cdots & 0\\ 0 & G_{32}-\frac{\lambda_{3}\left(k\right)}{2} & G_{33}-\lambda_{3}\left(k\right) & \cdots & 0\\ \vdots & \vdots & \vdots & \ddots & \\ 0 & 0 & 0 & \; & G_{mm}+\lambda_{m}\left(k\right) \end{array}\right] \end{array} \end{array}$$
here,
(14)$$\begin{array}{c} \lambda_{2r-1}\left(k\right)=\left(G_{r\;r}-p_{0}\right)\times U_{k}\;\;\\ \lambda_{2r}\left(k\right)=G_{\left(r+1\right)\;\;r}\times U_{k}\;\;\;\;\; \end{array}r=1,2,\cdots ,m/2$$
where $p_{0}$ is the system inertia factor, $U_{k}$ is the target threat value, $\lambda_{j}\left(k\right)$ is used to adjust the probability of the Markov transition according to the target threat risk value.


(3)Loss relating to target threat risk μ


As is known, the loss of the detection system caused by the target attack is mainly determined by the threat capability of the target. Therefore, we give the vector $\Gamma =\left\{\Gamma_{1},\Gamma_{2},\Gamma_{3}\right\}$ to describe the threat capability of the target according to prior experiences, in which $\Gamma_{1}$, $\Gamma_{2}$ and $\Gamma_{3}$ are the airborne weapons, jamming capability and category of the target, respectively. Additionally, we map elements of Γ into the threat loss domain by using the normalized loss function, and then the loss of the *j*-th threat level is achieved by the following form:(15)μj=fΓ,γ=∑i=13γi·exp−Γi−j−1/22−Γi−j−1/222δj22δj2
where $\gamma_{i}$ is the weight corresponding to $\Gamma_{i}$. $\delta_{j}$ is the standard variance of the *j*-th threat level.

## 4. Multi-Sensor Scheduling Based on Joint Risk Assessment

It is important to reasonably assign sensors to minimize potential risks for sensor scheduling [20,21]. Considering target loss risk and target threat risk, we propose a new scheduling method to assign sensors, ensuring risk assessment minimization.

### 4.1. Flow Chart of Proposed Method

Considering a variety risks, we propose the multi-sensor scheduling method, and give a joint risk assessment model with variable weight, multi-step prediction model of joint risk, multi-sensor scheduling optimization method, and so on. The flow chart of the proposed method is shown in Figure 3.

In this figure, the risk of target loss and the target threat risk can be calculated by using the detection performance of sensors, the sensor measurement information, and prior experiences, respectively. Moreover, the variable weights of all risks were adaptively given to construct the joint risk assessment. Additionally, the multi-sensor scheduling optimization model was constructed by minimizing the multi-step prediction of joint risk, and then the scheme of sensor scheduling was solved by using the convex relaxation method. More details related to the proposed method are shown as follows.

### 4.2. Joint Risk Assessment with Variable Weight

Supposing *N* sensors to track *M* targets (N>M), the joint risk assessment is:(16)$$J_{k}^{i,j}\left(W_{k},Q_{k}\right)=\alpha \left(k\right)W_{k}^{i,j}+\beta \left(k\right)Q_{k}^{i,j}$$
where $J_{k}^{i,j}$, $W_{k}^{i,j}$ and $Q_{k}^{i,j}$ are joint risk, target loss risk and target threat risk of the *i*-th sensor w.r.t *j*-th target, respectively. $\alpha \left(k\right)$ and $\beta \left(k\right)$ are adaptive risk weight about the target loss risk and the target threat risk, respectively.

The risks change with target moving, the adaptive weights $\alpha \left(k\right),\beta \left(k\right)$ related to risks are defined as:(17)$$\alpha \left(k\right)=\frac{\theta_{1}e^{-\frac{1}{2\sigma^{2}}{\left(d(k)-D(k)\right)}^{2}}}{\left(\theta_{1}e^{-\frac{1}{2\sigma^{2}}{\left(d(k)-D(k)\right)}^{2}}+\theta_{2}e^{-\frac{1}{2\sigma^{2}}{\left(d(k)-D(k)\right)}^{2}}\right)}$$
(18)$$\beta \left(k\right)=\frac{\theta_{2}e^{-\frac{1}{2\sigma^{2}}{\left(d(k)-D(k)\right)}^{2}}}{\left(\theta_{1}e^{-\frac{1}{2\sigma^{2}}{\left(d(k)-D(k)\right)}^{2}}+\theta_{2}e^{-\frac{1}{2\sigma^{2}}{\left(d(k)-D(k)\right)}^{2}}\right)}$$
where $\sigma^{2}$ is the noise variance of sensor measurement. $d\left(k\right)$ is the distance between a target and a sensor, and D(k) is the average distance between this target and all sensors. $\theta_{1}$ and $\theta_{2}$ are parameters related to loss risk and threat risk, respectively. Obviously, adaptive weights $\alpha \left(k\right)$ and $\beta \left(k\right)$ also change with distance $d\left(k\right)$.

### 4.3. Multi-Step Prediction Model of Joint Risk Assessment

To more efficiently design the scheme of sensor scheduling, risk prediction is necessary according to the current target state. Moreover, multi-step prediction of risk more efficiently ensures security of the detection system more than that of the single-step risk prediction.

Here, we propose a multi-step prediction model of joint risk as:(19)$$\begin{array}{c} J_{k+H}^{i,j}\left(W_{k},Q_{k}\right)=\left[\alpha \left(k\right)W_{k}^{i,j}+\beta \left(k\right)Q_{k}^{i,j}\right]+E\left\{\sum_{h=2}^{H}\left[\alpha \left(k\right)W_{k+h-1}^{i,j}+\beta \left(k\right)Q_{k+h-1}^{i,j}\right]\right\} \end{array}$$
where $J_{k+H}^{i,j}\left(W_{k},Q_{k}\right)$ is the cumulative joint risk from moment to moment k+H−1 (*H* is the number of steps); also, (i,j) expresses the *i*-th sensor tracking the *j*-th target.

### 4.4. Multi-Sensor Scheduling Optimization

To make the scheme of sensor scheduling, a multi-sensor scheduling problem considering joint risk prediction can be constructed as:(20)$$\begin{array}{c} \begin{array}{c} \underset{\psi_{k}^{i,j}}{\mathrm{minimize}}\sum_{i=1}^{N}\sum_{j=1}^{M}\left\{\psi_{k}^{i,j}\cdot \left\{\left[\alpha \left(k\right)W_{k}^{i,j}+\beta \left(k\right)Q_{k}^{i,j}\right]+E\left\{\sum_{h=2}^{H}\left[\alpha \left(k\right)W_{k+h-1}^{i,j}+\beta \left(k\right)Q_{k+h-1}^{i,j}\right]\right\}\right\}\right\}\\ subject\;to\;\;\sum_{i=1}^{N}\psi_{k}^{i,j}\geq 1\\ \;\;\;\;\;\;\;\;\;\;\;\;\;\;\;\;\;\;\;\sum_{j=1}^{M}\psi_{k}^{i,j}\geq 1\\ \;\;\;\;\;\;\;\;\;\;\;\;\;\;\;\;\;\;\psi_{k}^{i,j}\in \left\{0,1\right\},\;i=1,\cdots ,N\;\;\;j=1,\cdots ,M \end{array} \end{array}$$
where $\psi_{k}$ with size N×M is the scheduling scheme of sensor management, and $\psi_{k}^{i,j}$ is the element of $\psi_{k}$ with *i*-th row and *j*-th column; also, $\psi_{k}^{i,j}=1$ expresses sensor *i* assigned to track target *j*.

Further, $\psi_{k}^{i,j}\in \left\{0,1\right\}$ in (20) is non-convex, leading to the problem being NP-hard. To cope with the problem, we relax $\psi_{k}^{i,j}\in \left\{0,1\right\}$ into $\psi_{k}^{i,j}\in \left[0,1\right]$, and then a convex optimization problem is reformulated as: (21)$$\begin{array}{c} \begin{array}{c} \underset{\psi_{k}^{i,j}}{\mathrm{minimize}}\sum_{i=1}^{N}\sum_{j=1}^{M}\left\{\psi_{k}^{i,j}\cdot \left\{\left[\alpha \left(k\right)W_{k}^{i,j}+\beta \left(k\right)Q_{k}^{i,j}\right]+E\left\{\sum_{h=2}^{H}\left[\alpha \left(k\right)W_{k+h-1}^{i,j}+\beta \left(k\right)Q_{k+h-1}^{i,j}\right]\right\}\right\}\right\}\\ subject\;to\;\;\sum_{i=1}^{N}\psi_{k}^{i,j}\geq 1\\ \;\;\;\;\;\;\;\;\;\;\;\;\;\;\;\;\;\;\;\sum_{j=1}^{M}\psi_{k}^{i,j}\geq 1\\ \;\;\;\;\;\;\;\;\;\;\;\;\;\;\;\;\;\;0\leq \psi_{k}^{i,j}\leq 1,\;i=1,\cdots ,N\;\;\;j=1,\cdots ,M \end{array} \end{array}$$

Given the fact that the solution of (21) is not a direct solution of problem (20), to this end, we employ the discretization and legalization method (D-L) to handle [8], where the details are in Table 2.

### 4.5. Algorithm Implementation

According to the above analysis, more details related to the proposed multi-sensor scheduling algorithm, in which joint risk assessment with variable weight is considered, and the pseudocode is shown in Table 3.

## 5. Simulation and Analysis

### 5.1. Scene and Parameters

To evaluate the effectiveness of the proposed scheduling algorithm (PMA), three existing approaches are selected to compare with the PMA algorithm, in which the characteristics are as follows:(1)The random scheduling approach (RMA) [14]: it randomly assigns sensors to detect and track targets, leading to its main application in the case where unknown variables cannot be solved immediately.(2)The closest scheduling approach (CMA) [15]: it selects sensors closer to the target to detect and track targets. Therefore, the better estimation accuracy of the target state can be obtained.(3)The myopic scheduling approach (MMA) [16]: it regards minimization of one-step joint risk prediction as the optimization rule of multi-sensor scheduling, in which schemes of sensor scheduling are obtained according ahead of time.

In this paper, the target state model and sensor measurement model are (1) and (2), respectively. The parameters of four sensors (S1,S2,S3,S4) are shown in Table 4, which are employed to track the moving target. The initial state estimations of three targets are position (50,3,0.4)km, velocity (−0.005,−0.002,0)km/s and acceleration $\left(-0.0001,0,0\right)\;\mathrm{km}/s^{2}$, respectively. The parameters related to loss risk and threat risk are $\theta_{1}=0.15,\theta_{2}=0.4$. The level of target threat is m=2, the initial target threat probability is p=[0.9,0.1], the initial Markov transition probability is $G_{ij}=\left[0.9,0.1;0.1,0.9\right]$. At the same time, simulation time is 1200 s.

Here, simulation software is MATLAB R2016a, and configuration of computer core hardware: CPU Inter(R) Core(TM) i5-9500U 3.00 GHz, 16 G memory.

### 5.2. Results and Analysis

The efficiency of risk control and the accuracy of risk prediction are verified in this section. Furthermore, the performance of PMA and the comparison with three other methods are simulated and analyzed as follows.

#### 5.2.1. Simulation of PMA

(1) Determine the number of steps *H*

Calculation accuracy and time complexity are influenced by the number of steps *H* in (19). According to analysis, the larger the number of steps *H*, the more the cumulative joint risk, the better the multi-step risk prediction, but the time complexity increases. To this end, it is important to select the right number of steps *H* for balancing time complexity and calculation accuracy. The change of the cumulative joint risk with number of steps *H* is shown in Figure 4.

In Figure 4, it is obvious that the cumulative joint risk decreases with the increase of the number of steps *H*. It can be found that the cumulative joint risk happens with a small change from step *H* = 5 to *H* = 8, meaning fewer optimizations in the sensor scheduling scheme. Considering time complexity increases with a bigger number of steps *H*, this paper takes the number of steps *H* = 5.


(2)Solution of joint risk


The joint risk assessment changes with a target move. The joint risk comparison of the real and prediction value is shown in Figure 5.

In Figure 5, the prediction risk and the real risk describe the change of joint risk. In Figure 5a,b, symbol “A” expresses the same turning point under two different physical indicators (time, distance). Moreover, before “A”, there is a greater distance between sensors and the target is longer, as expected. The loss risk of the target is the main factor of the joint risk. However, with the distance between sensor and target gradually decreasing, the loss risk probability of the target decreases, resulting in a downward trend of the joint risk. Similarly, after “A”, the loss risk probability of the target decreases too, while the threat risk of the target is the main factor of the joint risk. Additionally, the threat risk probability of the target keeps increasing, leading to an increasing trend of the joint risk. It can be seen that the curves of risk prediction and real risk are roughly the same, it can reasonably explain the risk assessment problems; moreover, it can verify the accuracy of risk prediction.


(3)Scheme of sensor scheduling


Through the convex optimization, the minimization problem of joint risk is solved; however, the solution is a discrete 0–1 scheme. To cope with the problem, the D-L method is necessary to obtain an optimal scheme of sensor scheduling. As an example, the sensor scheduling matrix is discretized by threshold by using the D-L method as:

It can be seen in Figure 6, the elements in the matrix that are greater than or equal to the threshold are set to 1, otherwise 0. Therefore, a continuity matrix is transformed into a discrete 0–1 matrix. According to the given constraint, in which a sensor can only track one target and a target can only be tracked by one sensor, the sensor scheduling matrix is also legitimized. Since the elements of the first and second row in the first column are both 1, and the corresponding optimization value 0.6<0.8, the first-row element in the first column is set to 0, so as to complete the legalization.

After D-L processing, the sensor scheduling scheme can be obtained by solving the optimization problem. As expected, some sensors can be scheduled to track the target scheme as shown in Figure 7.

In this figure, taking the period from 40 s to 50 s as an example, during the moment of 40–41 s, 42–43 s, 44–45 s, 46–47 s and 48–50 s, S1, S4, S1, S2 and S3 were assigned to detect and track target, respectively. Indeed, the above scheme of sensor scheduling can be obtained in real time by solving the minimum multi-step prediction joint risk problem, leading to reduced joint risk of the detection system.

#### 5.2.2. Comparison of Different Methods


(1)Joint risk comparison


Comparing the proposed method PMA with the other three methods CMA, RMA and MMA, we analyze their joint risk assessment, which can be seen in Figure 8 and Figure 9. Here, Figure 8 is the joint risk comparison of four methods, and Figure 9 is the comparison of cumulative joint risk, cumulative threat risk and cumulative loss risk.

In Figure 8, the cumulative risk of PMA is lower than other methods (RMA, CMA and MMA) during the total simulation time. Taking the moment 60 s as an example, compared with methods RMA, CMA and MMA, the joint risk of method PMA reduces by 30.43%, 11.24% and 7.18%, respectively. Moreover, it can be seen from Figure 9, compared with three methods, the cumulative joint risk of method PMA is reduced by 10.05%, 12.96% and 3.14%, the cumulative threat risk reduced by 6.63%, 9.21% and 1.95%, the cumulative loss risk reduced by 47.81%, 56.57% and 20.34%, respectively. The reason behind this observation is that the method RMA randomly assigns sensors. The risk prediction model is not considered to control the risk, leading to the highest risk. Although the CMA method can guarantee better sensor measurement information, it has high threat risk. Moreover, the risk prediction model is also not considered, the resulting risk is controlled with delay. The MMA method considers risk as a one-step prediction model to better predict and control risk, but it is not considered to estimate risk, leading to the best risk control effect hardly being obtained. According to the above analysis, compared with methods RMA, CMA and MMA, the PMA method can better balance the target threat risk and target loss risk, controlling joint risk and improving system safety.

To better illustrate the advantage of the PMA method, Figure 10 shows the cumulative risk comparison of four methods on the joint risk, loss risk and threat risk.

As expected, the PMA method has the smallest cumulative joint risk, cumulative loss risk and cumulative threat risk in Figure 10. For example, at moment 60 s, compared with methods RMA, CMA and MMA, the cumulative loss risk of PMA reduces by 68.29%, 43.72% and 11.59%, respectively. Moreover, the cumulative threat risk of PMA respectively reduces by 11.23%, 6.72% and 1.91%, and the cumulative joint risk of PMA respectively reduces by 17.29%, 11.32% and 3.73%. These results show that the PMA method is effective in risk assessment, controlling the joint risk and providing a better scheme of sensor scheduling.


(2)Tracking error comparison


Obviously, the tracking error of the sensor is one important indicator to evaluate the scheme of sensor scheduling, i.e., the smaller the tracking error, the higher the tracking accuracy of the sensor. We regard the maximum tracking error (MTE) and average tracking error (ATE) as indicators to evaluate four methods, which can be seen in Table 5.

It can be seen from this table, compared with the other three methods, the proposed PMA method is not minimum at MTE. However, compared with RMA, CMA and MMA, the average tracking error of PMA reduces by 51.9%, 7.4% and 22.2%, respectively. Moreover, compared with the RMA method, the PMA method decreased by 34.1% with respect to ATE. These results illustrate that the PMA method does not reduce target tracking accuracy, while emphasizing risk control.


(3)Complexity analysis


The complexity of the proposed PMA method is mainly determined by the number of sensors and targets. The optimization problem (21) contains *N* sensors and *M* targets, and the complexity of multi-sensor scheduling by convex optimization theory is $O\left(\sqrt{N\times M}\left(N\times M\right)\right)$. Since the CMA and MMA methods use exhaustive methods to solve sensor scheduling, their complexity both are $O\left({\left(N\times M\right)}^{2}\right)$. Therefore, the complexity of the PMA method is significantly lower than that of CMA and MMA. Furthermore, the RMA method randomly schedules sensor resources, in which the optimization problem of multi-sensor scheduling is not involved, leading to complexity, is the lowest. However, it can be analyzed from (1) and (2) mentioned above that compared with RMA, PMA has a great improvement in risk control and target tracking accuracy. For comparison, the complexities of the proposed PMA method and the other three methods are shown in Table 6.

## 6. Conclusions

In this paper, a multi-sensor scheduling method based on joint risk assessment is presented. We combine target loss risk and target threat risk by variable weights to construct a joint risk model. We employ the multi-sensor scheduling problem minimizing multi-step joint risk prediction, by which the suboptimal scheme of sensor scheduling and more accurate prediction of joint risk can be achieved, leading to reduced potential task risk caused by the sensor tracking target. In our future work, it would be interesting to further narrow the performance gap between the approximate solution and the optimal solution of the original optimization problem in (20).

## Figures and Tables

**Figure 1 entropy-24-01315-f001:**
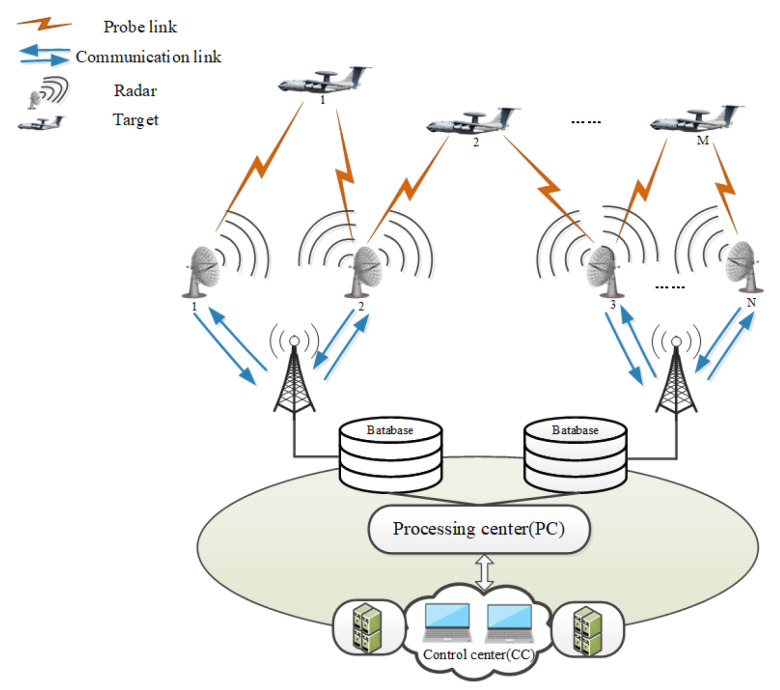
Multi-sensor cooperative detection system.

**Figure 2 entropy-24-01315-f002:**
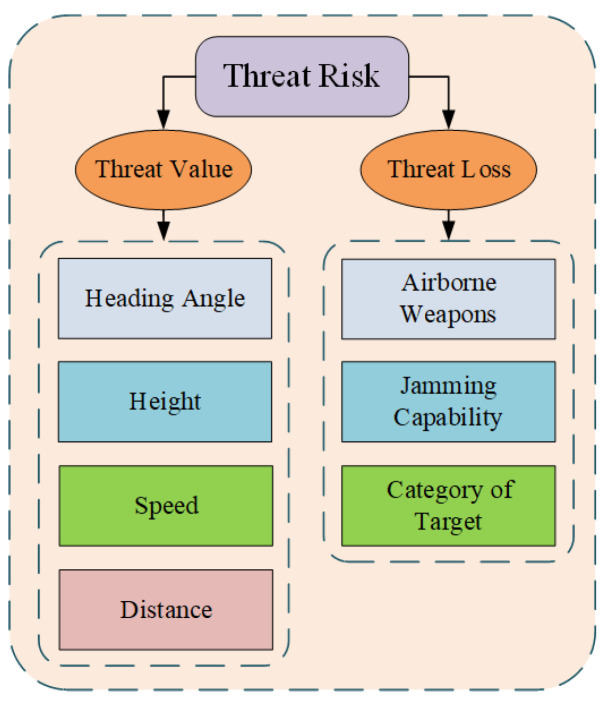
Composition of target threat risk.

**Figure 3 entropy-24-01315-f003:**
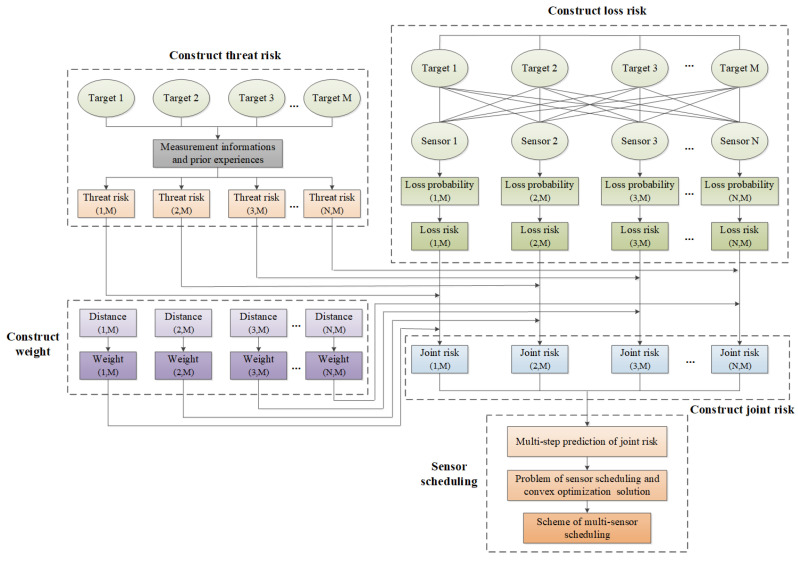
Flow chart of the proposed method.

**Figure 4 entropy-24-01315-f004:**
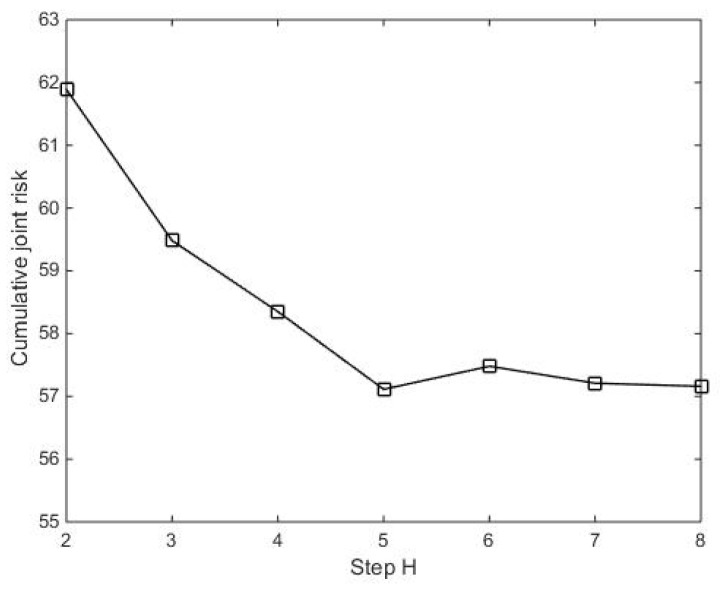
The cumulative joint risk change.

**Figure 5 entropy-24-01315-f005:**
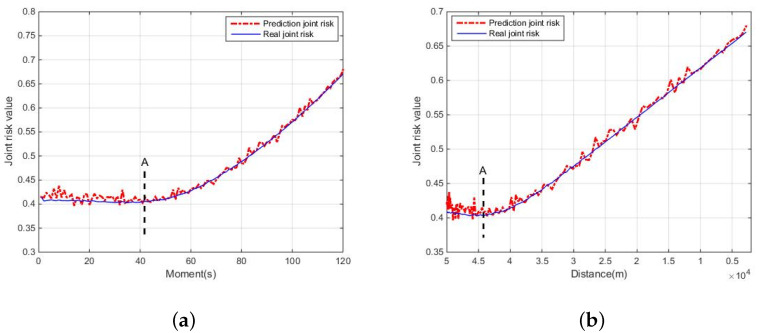
Comparison of real and joint prediction. (**a**) Change with moment; (**b**) change with distance.

**Figure 6 entropy-24-01315-f006:**
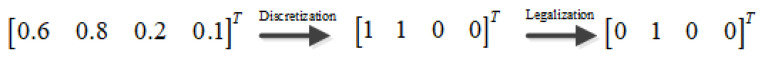
An example of D-L.

**Figure 7 entropy-24-01315-f007:**
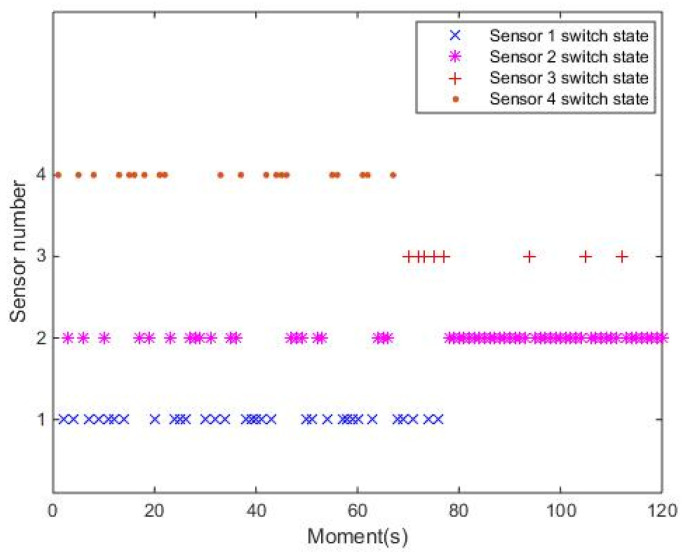
Scheme of sensor scheduling.

**Figure 8 entropy-24-01315-f008:**
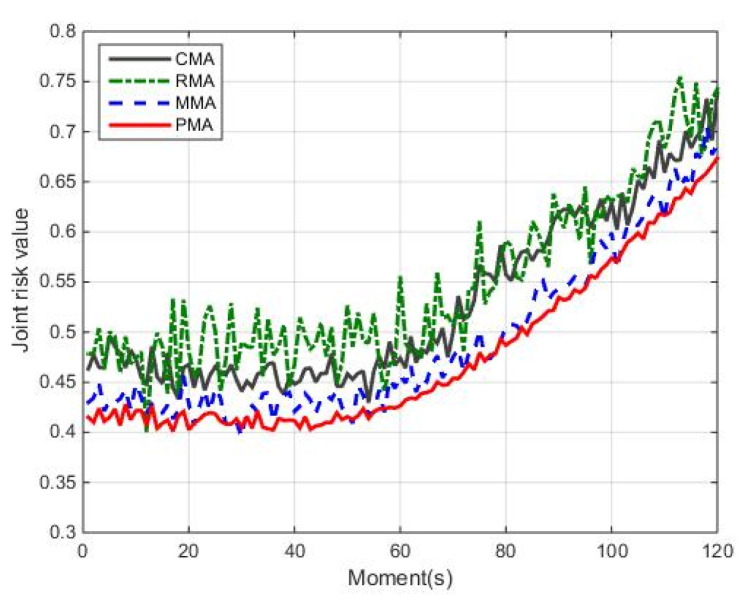
Different joint risks of four methods.

**Figure 9 entropy-24-01315-f009:**
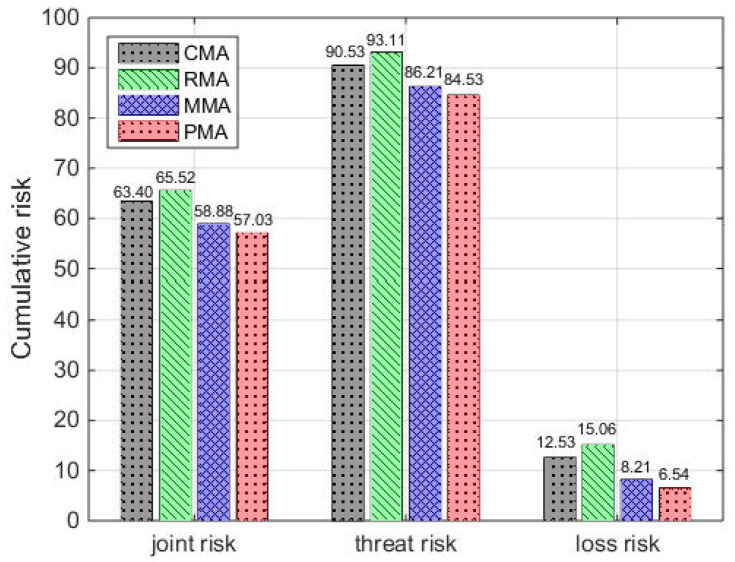
Cumulative risk of four methods.

**Figure 10 entropy-24-01315-f010:**
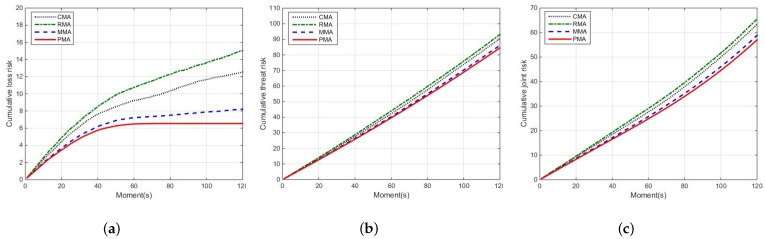
The cumulative risk comparison of four methods. (**a**) The cumulative loss risk; (**b**) the cumulative threat risk; (**c**) the cumulative joint risk.

**Table 1 entropy-24-01315-t001:** Normalized function.

Continuous Variables	Normalized Functions	Domain
Heading angle	$C=\exp\left(-{\left[f_{C}^{}\left(U_{}^{C}\right)\right]}^{2}\right)$	$0^{\circ }\leq U^{C}\leq 180^{\circ }$
Height	$A=1-\exp\left({\left[f_{A}^{}\left(U_{}^{A}-U_{\max}^{A}\right)\right]}^{2}\right)$	$0<U^{A}\leq U_{\max}^{A}$
Speed	$V=\exp\left({\left[f_{V}^{}\left(U_{}^{V}-U_{\max}^{V}\right)\right]}^{2}\right)$	$0<U^{V}\leq U_{\max}^{V}$
Distance	$R=1-\exp\left({\left[f_{R}^{}\left(U_{}^{R}-U_{\max}^{R}\right)\right]}^{2}\right)$	$0<U^{R}\leq U_{\max}^{R}$

**Table 2 entropy-24-01315-t002:** Multi-sensor scheduling based on joint risk prediction (SMBRP).

**Input** Multi-step risk prediction $J_{k+H}^{i,j}\left(W_{k},Q_{k}\right)$
**Output** Scheme of sensor scheduling $\psi_{k}^{*}$
(1) Initializing parameters;
(2) Constructing multi-sensor scheduling optimization problem based on joint risk prediction is
according to (20);
(3) Relaxing $\psi_{k}^{i,j}\in \left\{0,1\right\}$ into $\psi_{k}^{i,j}\in \left[0,1\right]$, then getting (21);
(4) Calculating $\psi_{k}$ according to (21);
(5) Computing $\psi_{k}$ by using D-L method and getting suboptimal scheduling scheme $\psi_{k}^{*}$ of (20).

**Table 3 entropy-24-01315-t003:** Algorithm pseudocode.

**Input** Sensors measurements $z_{k}$, target threat capability Γ
**Output** Scheme of sensor scheduling $\psi_{k}^{*}$
(1) Initializing parameters;
(2) **For** $k=1:k_{\max}$ **do**
(3) Calculating the target loss risk $W_{k}$ according to (3);
(4) Calculating the target threat risk $Q_{k}$ according to (10);
(5) Calculating the adaptive weight of target loss risk and target threat risk α(k),β(k) based on (17) and (18);
(6) Calculating the joint risk $J_{k}^{i,j}\left(W_{k},Q_{k}\right)$ according to (16);
(7) Calculating the multi-step prediction of joint risk $J_{k+H}^{i,j}\left(W_{k},Q_{k}\right)$ by using (19);
(8) Constructing the optimization problem of joint risk-based multi-sensor scheduling according to (20);
(9) Using SMBRP to solve the optimization problem (21), and getting the scheme of sensor scheduling $\psi_{k}^{*}$;
(10) **End for**

Explain: The specific algorithm flow of SMBRP is shown in Table 2.

**Table 4 entropy-24-01315-t004:** Related parameters of sensors.

Sensors	Position/km	Detection Power	Sampling period/s	Measurement noise/km	False Alarm Probability
S1	(0,0,0)	24.4	10	0.61	$10^{-8}$
S2	(0.1,0,0)	25.4	10	0.60	$10^{-8}$
S3	(0,0.1,0)	24.8	10	0.59	$10^{-8}$
S4	(0.1,0.1,0)	26.1	10	0.60	$10^{-8}$

**Table 5 entropy-24-01315-t005:** Comparison of tracking error.

Method	MTE (km)	ATE (km)	The Percentage Reduction of RMA in ATE
CMA	0.50	0.29	29.3%
MMA	0.61	0.33	19.5%
RMA	0.69	0.41	-
PMA	0.52	0.27	34.1%

**Table 6 entropy-24-01315-t006:** Comparison of complexity.

Method	Complexity in General
CMA	$O\left({\left(N\times M\right)}^{2}\right)$
MMA	$O\left({\left(N\times M\right)}^{2}\right)$
RMA	$O\left(N\times M\right)$
PMA	$O\left(\sqrt{N\times M}\left(N\times M\right)\right)$

## Data Availability

Not applicable.

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
