# Peer review of "Multi-Sensor Scheduling Method Based on Joint Risk Assessment with Variable Weight"

_entropy, 2022, doi:10.3390/e24091315_

Round 1
Reviewer 1 Report
In this paper, a multi-sensor scheduling method based on joint risk assessment is presented, in which suboptimal scheme of sensor scheduling and more accurate prediction of joint risk can be achieved. The proposed method can effectively reduce the potential task risk caused, additionally, effectiveness of the proposed method is proved by simulation. Some comments are given as follows:
1. There are many sensor scheduling methods for multi-sensor cooperative system, but there are no relevant papers in recent years.
2. Since discrete risk and continuous risk cause risk assessment complexity, and this paper tries to solve these problems by proposed joint risk assessment model and multi-step joint risk prediction method, the performance should be analyzed and compared.
3. In order to show the contribution of the paper, authors are suggested to more clearly highlight what the difficulties and challenges are for the problem concerned.
4. The novelty and original contribution of the paper is somewhat unclear. For example, page 2, main contributions of this paper are described, and the second innovation only give that the improved model avoid frequent switching of sensor working modes, however, it does not point out specific performance improvement. Thus, we suggest to add advantages of the model in this paper.
5. The advantages of the developed results should be further clearly shown.
6. Some descriptions are errors in this paper. For example, page 13, “MTE(km)m” in Table 5 has an extra “m”, is wrong. Authors are suggested to check and polish this paper again.
Author Response
请参阅附件。

Reviewer 2 Report
In this paper, a multi-sensor scheduling method based on joint risk assessment with variable weight is proposed to reduce target threat risk and target loss risk in multi-sensor cooperative detection system. The target threat risk and loss risk are described, the variable weight joint risk assessment model is given, and the corresponding sensor scheduling scheme is designed. The simulation results show that the proposed method is feasible and superior. This paper has a clear structure and rich content and the method is innovative and valuable in application.
Suggestions for this paper:
Description of target tracking method can be added.
In the simulation, the description of the actual motion parameters of the target can be improved.
